# Synthesis and Characterization of Novel Nanoporous Gl-POSS-Branched Polymeric Gas Separation Membranes

**DOI:** 10.3390/membranes10050110

**Published:** 2020-05-24

**Authors:** Ilnaz I. Zaripov, Ilsiya M. Davletbaeva, Zulfiya Z. Faizulina, Ruslan S. Davletbaev, Aidar T. Gubaidullin, Artem A. Atlaskin, Ilya V. Vorotyntsev

**Affiliations:** 1SIBUR LLC, 16, bld.3, Krzhizhanovskogo str., GSP-7, 117997 Moscow, Russia; zaripovilnaz@gmail.com; 2Department of Technology of Synthetic Rubber, Kazan National Research Technological University, 68 Karl Marks str., 420015 Kazan, Russia; faizulina.alin@yandex.ru; 3Department of Materials Science, Welding and Industrial Safety, Kazan National Research Technical University n.a. A.N. Tupolev— KAI, 10 Karl Marks str., 420111 Kazan, Russia; darus@rambler.ru; 4Arbuzov Institute of Organic and Physical Chemistry, FRC Kazan Scientific Center of Russian Academy of Science, 8 Arbuzov str., 420088 Kazan, Russia; aidar@iopc.ru; 5Laboratory of Membrane and Catalytic Processes, Nizhny Novgorod State Technical University, n.a. R.E. Alekseev, 24 Minin str., 603950 Nizhny Novgorod, Russia; atlaskin@gmail.com (A.A.A.); ilyavorotyntsev@gmail.com (I.V.V.); 6Department of Membrane Technology, Mendeleev University of Chemical Technology of Russia, Miusskaya Sq. 9, 125047 Moscow, Russia

**Keywords:** nanoporous polymers, supramolecular structure, gas transport membranes, gas separation

## Abstract

Novel nanoporous Gl-POSS-branched polymers based on the macroinitiator of anionic nature, 2,4-toluene diisocyanate, and octaglycidyl polyhedral oligomeric silsesquioxane (Gl-POSS) were obtained as gas separation membranes. The synthesis of polymers was carried out using various loads of Gl-POSS. It was found that the main reaction proceeding with 2,4-toluene diisocyanate is the polyaddition, accompanied by the isocyanate groups opening of the carbonyl part. This unusual opening of isocyanate groups leads to the formation of coplanar acetal nature polyisocyanates (O-polyisocyanate). The terminal O-polyisocyanate links initiate the subsequent opening of the epoxide rings in Gl-POSS. As a result, Gl-POSS serves as a hard and bulky branching agent and creates the specific framing supramolecular structure, which leads to the formation of nanopores in the polymer, where the flexible polyether components are located inside the cavities. Thermal, mechanical, physical, and chemical properties of the obtained polymers were studied at various Gl-POSS contents in the polymer matrix. It was found that these polymers show high selectivity of gas transport properties for pure ammonia relative to nitrogen and hydrogen at ambient temperature. Measurements showed that the gas permeability coefficients and the values of ideal selectivity were in a non-additive dependence to the Gl-POSS content.

## 1. Introduction

Throughout recent decades, greater attention has been turned toward the use of polymeric gas separation membranes in separation and purification processes. This is largely owing to their potential energy efficiency and minimal ecological impact as compared to traditional, but potentially dangerous, distillation, crystallization, and sorption methods [1,2,3]. Despite the advantages of membrane–gas separation, this method has limited application [4,5,6,7,8,9,10]. The reason is the relatively low mechanical resistance of polymeric membranes, which is stipulated by weak interactions between macromolecules, or the existence of a statistically distributed free volume, allowing gases to permeate through the polymer. Polymer membranes can be glassy or rubbery [1,2,3,4,5]. The formation of free volume in polymeric membranes is stipulated by the features of their macromolecular and supramolecular structure [11,12]. Glassy polymers are characterized by a rigid macromolecular structure, resulting in the free volume within the polymeric structure [11,12]. Most often, the rigid polymer backbone allows smaller molecules to permeate the membrane with less resistance than larger molecules [13]. This results in low permeability while maintaining high selectivity when using glassy polymeric membranes. A general rule that has emerged in the development of membrane technologies is that the modification of the glassy polymer leads to an improvement of one of these characteristics but to the deterioration of the other [13]. 

Conversely, gas permeation through rubbery polymers is due to weak interactions between the macromolecules. As a result, the permeability is higher as compared to glassy polymers, but the selectivity is significantly diminished [1,2,3,4,5]. One of the greatest challenges in this work is the development of new polymeric gas separation membranes able to achieve high values of permeability and selectivity using a single material. One possible approach to achieve this goal is the synthesis of block copolymers capable of self-assembly in a variety of ordered nanosizes and domain structures [14,15]. The synthesis of amphiphilic block-copolymers allows the combination of these factors, giving rise to remarkable polymers with desirable gas permeability properties [16,17,18]. Features of their structure can combine segments, providing high sorption properties of the surface, high free volume, and weak intermolecular interactions. As the self-organization is the driving force when forming the polymer matrix of the block-copolymers, it is not always possible to attain other desirable properties, such as mechanical strength and heat resistance.

Another possible improvement of gas separation membrane properties (i.e., selectivity and permeability) is active transport, especially in the case of the separation of nonpolar gases, and active penetrants like carbon dioxide [19,20,21], hydrogen sulfide [22,23], and ammonia [24,25,26,27,28]. However, the implementation of the membrane separation process for the aforementioned gases encounters another problem: Chemical stability of a polymer arising from its operation time. The existing experimental data provides poor information about the ammonia permeability, despite ammonia having the largest production capacity in the world. Moreover, the ammonia permeation mechanism remains elusive even today. Membrane [26] based on vinylidene fluoride and tetrafluoroethylene has high selectivity to ammonia/hydrogen, which is equal to 80 at rather high permeability to ammonia, which is realized due to the possible interaction of ammonia with the polymer structure, leading to membrane degradation and recoloring. It is known that polyvinylidene fluoride is destroyed by prolonged contact at ambient temperature with aniline, which is a weaker base than ammonia. This fact provides some level of predictive power of the destruction of the copolymer with time. 

The membrane based on copolymers of silicon rubber with polyethylene glycol demonstrates a rather high ammonia steady state permeability value (2.24 10^−4^ cm^3^/cm^2^ sec cm Hg) [29]. Effective separation of ammonia from hydrogen could be achieved with the help of the porous media (ceramic or metal) with melted salt, such as ZnCl_2_, with which ammonia can react [30], especially at high temperatures in the range of 523–623 K. One should note that the ammonia production process, invented and industrially implemented by Haber and Bosch, has remained largely unchanged for nearly a century. Certainly, in due course, process optimization must be accomplished, and membrane gas separation is one of the most promising ways to improve this classical process [24,25,31,32,33,34]. Given this need, ammonia permeation served as one facet of the inspiration for this research.

Another approach utilizes hybrid materials based on room temperature ionic liquids (RTILs), which can be used as selective media or additives to the polymeric matrix [35,36,37,38,39]. A number of works concerning carbon dioxide separation by RTILs have been published for the last decade. In the majority of these studies, ionic liquids with quaternary cations or protonated imidazole were used owing to their high permeability (about 1000 Barrer) and selectivity for carbon dioxide/nitrogen of more than 50. 

However, there is a greater possibility to improve the polymeric membrane properties by the influence on the polymer matrix structure—to create block-structured organic-inorganic polymers [40,41]. In such hybrid polymers, inorganic components enable the formation of mechanically robust frameworks as well as nanopores with a size less than 2 nm. These pores result in increased permeability due to the increased contribution of diffusion components. The flexible segments (polyoxyethylene component being the most frequently used) in the composition of such polymers leads to an increased solubility of polar molecules (ammonia etc.) due to the presence of native hydrogen bonding sites. This fact is a prerequisite for significant growth in the selectivity of gas separation processes. 

The ability of using organic-inorganic hybrid materials for gas separation arises from the synergizing effect of both organic and inorganic components [42,43,44]. Currently, there is a number of works that have demonstrated that octafunctionalized polyhedral oligomeric silsesquioxanes (POSSs) offer an efficient route for the development of novel hybrid nanocomposites [45,46,47,48,49]. Now POSS is widely used to obtain polymeric nanocomposites because of the improving mechanical properties [50,51], increasing glass-transition temperature [52], and increasing homopolymer blend toughness. Moreover, the versatile chemistry of POSS lends itself to nearly infinite chemical modifications. The cubic silsesquioxane unit is a well-defined nanometer-sized structure (hollow silica core with a nanopore diameter of about 0.3–0.4 nm [53]; the size of POSS is around 1–3 nm) with a high surface area, controlled porosity, and various functionalities. Furthermore, these units can be tailored with different reactive or non-reactive organic groups at the tetravalent Si atoms. One possible approach is copolymerization of functionalized POSSs with a conventional polymer. On the other hand, reacting functionalized POSS with a reactive polymer provides another approach to build up polymer-tethered POSS nanocomposites [54]. The smaller size and tailorable organic groups make POSS an attractive and promising material for gas separation applications [42,45,46,47]. POSS-based nanocomposites as gas separation membranes were obtained with Matrimid [45], polystyrene [42], polyetheramine [46], some fluorine-containing polyimides [47], and PEBAX MH 1657 [48]. Additionally, a number of POSS-based composites proposed for other membrane processes have been offered in the literature [49]. This study discusses the permeability of some nonpolar gases and ammonia. As it was already found, a polymer based on aromatic isocyanates can be used as an ammonia resistant separation membrane [41]. In [55,56], the possibility of epoxide ring opening in octaglycidyl polyhedral oligomeric silsesquioxane (Gl-POSS) by both anionic and cationic mechanisms was established. According to these studies, polyaddition of epoxides proceeds at 353–413 K by both anionic and cationic mechanisms. Gl-POSS-based hybrid homopolymers and copolymers obtained using thermal cationic polymerization initiated by diaryl iodonium fluoride-borate were studied [57]. The homopolymer of Gl-POSS and copolymer of Gl-POSS with three glycidyloxypropyl trimethoxysilanes were synthesized through thermal cationic ring-opening polymerization using a diaryliodonium fluoride-borate initiator and benzoyl peroxide as a co-initiator. 

In our previous studies [53], block-copolymers (POI-0) were obtained by polyaddition of tolylene-2,4-diisocyanate (TDI) to a macroinitiator possessing an anionic nature (MI), which is a potassium-substituted amphiphilic block-copolymer of propylene oxide and ethylene oxide (PPEG). The acetal nature polyisocyanate (O-polyisocyanate) formed owing to the reaction conditions, which resulted in the exposure of isocyanate groups on the carbonyl component [54,58]. These O-polyisocyanate blocks have a coplanar trans-configuration. 

The previous works led to the present study on the synthesis and characterization of the novel polymers with octaglycidyl polyhedral oligomeric silsesquioxane as the branching centers of O-polyisocyanates, which are an integral part of the amphiphilic block copolymers, followed by hybrid copolymers’ formation for the purpose of active penetrant gas separation, namely ammonia.

## 2. Materials and Methods

### 2.1. Materials

Block-copolymer of propylene oxide with ethylene oxide (PPEG) with formula HO[CH_2_CH_2_O]_n_[CH_2_(CH_3_)CH_2_O]_m_[CH_2_CH_2_O]_n_K, where n ≈ 14 and m ≈ 48, molecular weight 4200, content of potassium alcoholate groups is 10.9 wt% from the total the number of functional groups, was purchased from PJSC Nizhnekamskneftekhim (Nizhnekamsk, Russia). 2,4-toluene diisocyanate 98% (TDI) was purchased from Sigma-Aldrich. Octaglycidyl polyhedral oligomeric silsesquioxane (Gl-POSS) was purchased from Hybrid Plastics (Hattiesburg, MS, USA). Gl-POSS is unstable over time and thus was closed tightly and stored in a cold dry place at 283 K. The reaction media for the synthesis of block-copolymers was ethyl acetate. PPEG was additionally dried at reduced pressure (approximately 0.07 kPa) and at an elevated temperature of 353 K down to a 0.01 wt% moisture concentration. 

In the gas separation investigation of prepared polymers, single gases (hydrogen, nitrogen, and ammonia) were used. Hydrogen and nitrogen with a purity no less than 99.995% were purchased from NII KM, (Moscow, Russia). High-purity ammonia 99.99999% from Firm HORST Ltd. (Moscow, Russia) was employed for permeability studies.

### 2.2. Synthesis of Polymers Based on TDI, PPEG, and Gl-POSS (POI-Gl-POSS) 

The reaction was carried out in ethyl acetate at 25 °C in a flask equipped with a reflux condenser. The polymerization process proceeded with constant stirring using a magnetic stirrer. The flask contained PPEG (1 g) and ethyl acetate (4.86 g). The calculated amount of Gl-POSS as a curing agent was added to PPEG solution and under constant stirring. The reaction mass was mixed at a given temperature until the complete dissolution of Gl-POSS took place. After this, TDI (0.62 g) at a molar ratio of [PPEG]:[TDI] = 1:15 was introduced. The Gl-POSS content varied in the range of 0.0 to 20 wt%, and depending on this, the polymer was designated as POI-0, POI-Gl-POSS-0.1, POI-Gl-POSS-0.4, POI-Gl-POSS-0.5, POI-Gl-POSS-1, POI-Gl-POSS-2, POI-Gl-POSS-5, POI-Gl-POSS-8, POI-Gl-POSS-10, POI-Gl-POSS-15, and POI-Gl-POSS-20. The resulting solution of the polymer-forming system was cast in a Petri dish and cured at ambient temperature for 72 h. The ratio of [~O^−^ K^+^] (g/eq) and [
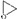
] groups (E/K) was taken into account, when calculating the content of Gl-POSS. Table 1 shows the ratio of these groups depending on the content of Gl-POSS. The calculation was performed on 100 g of PPEG. 

All samples were seasoned at room temperature at least 3 weeks for removing the solvent residuals before performing mechanical, physical, and gas transport tests.

### 2.3. Polymer Characterization

#### 2.3.1. Tensile Stress–Strain Measurements

Tensile stress–strain measurements were obtained from the film samples of size 40 mm × 15 mm with Universal Testing Machine Inspekt mini (Hegewald & Peschke Meß- und Prüftechnik GmbH, Nossen, Germany) at 293 ± 2 K, 1 kN. The crosshead speed was set at 50 mm/min and the test continued until sample failure. A minimum of five tests were analyzed for each sample and the average values were reported. 

#### 2.3.2. Infrared Spectroscopy Analysis

The Fourier transform infrared (FTIR) spectra of the products were recorded on an InfraLUM FT 08 Fourier transform spectrometer (Lumex, St. Petersburg, Russia) using the attenuated total reflection technique. The spectral resolution was 2 cm^−1^, and the number of scans was 60.

#### 2.3.3. Thermomechanical Analysis

The thermomechanical curves of polymer samples were obtained using a TMA 402 F (Netzsch, Selb, Germany) thermomechanical analyzer in the compression mode. The thickness of the sample was 2 mm, the rate of heating was 3 K/min from room temperature to 540 K in static mode, and the load was 2 N. 

#### 2.3.4. Mechanical Loss Tangent Measurements (MLT)

The MLT curves of polymer samples were taken using the dynamic mechanical analyzer DMA 242 (Netzsch, Selb, Germany) in the mode of the oscillating load. The force and stress–stain correspondence were calibrated using a standard mass. The thickness of the sample was 2 mm. Viscoelastic properties were measured under nitrogen. The samples were heated from room temperature to 540 K at the rate of 3 K/min and a frequency of 1 Hz. The mechanical loss tangent was defined as the ratio of the viscosity modulus G″ to the elasticity modulus G’.

#### 2.3.5. Thermal Gravimetric Analysis

Thermal gravimetric analysis (TGA) was performed using a STA-600 TGA–DTA combined thermal analyzer (Perkin Elmer, Waltham, MA, USA). The samples (0.1 g) were loaded in alumina pans and heated from 297 to 870 K at a rate of 5 K/min in a nitrogen atmosphere. 

#### 2.3.6. Dielectric Loss Tangents Measurements

The temperature dependence of the dielectric loss tangent (DLT) of polymer samples (thickness of 0.5–0.7 mm) was registered in the temperature range from 130 to 370 K at a frequency of 1 kHz. A facility consisted of the measuring cell that was placed in a Dewar vessel filled with nitrogen and to which an E7-20 RLC-meter and a B7-78 universal voltmeter functioning as a precision thermometer were connected.

#### 2.3.7. Atomic Force Microscopy Topology Analysis

An atomic force microscope (MultiMode V, USA) was used to reveal the morphology of the samples. The 250–350 kHz cantilevers (Veeco, Plainview, NY, USA) with silicone tips (tip curvature radius is of 10–13 nm) were used in all measurements. The microscopic images were obtained with a 512 × 512 resolution. The scanning rate was 1 Hz. Samples were fixed on a 12-mm metal SPM specimen discs using adhesive carbon tabs (AgarScientific, Essex, UK).

#### 2.3.8. Contact Angle Measurements

The angle of capillarity is a characteristic of the hydrophilicity (hydrophobicity) of the membrane surface. It is defined as the angle between the tangent drawn to the surface of the wetting liquid and the wetted surface of the solid, while always counting from the tangent to the side of the liquid phase. The tangent is drawn through the point of the three-phase contact: Solid phase (membrane), liquid (distilled water), and gas (air). Measurements were performed using an eyepiece micrometer. The measuring cell is mounted on a holder table. The test membrane is placed on a stand in the measuring cell and the illuminator lamp is turned on. Then, 20 mL of the test liquid is poured into the cuvette so that the liquid level is equal to half the height of the stand in the measuring cell. A drop of this liquid is carefully applied (1 mm^3^) to the surface of the membrane using a microsyringe. Adjust the position of the holder to obtain the sharpest image of the drop and plate. The parameters of the height h and the diameter d of the droplet are determined using an eyepiece micrometer. The drop is positioned so that it is located in the field of the eyepiece view between divisions 0 and 10. The image of a drop on the membrane is obtained by an inverted image. To measure the diameter of the drop base, the eyepiece-micrometer droplets rotate and move its crosshair vertically and combine it with the image of the plate (with the drop interface—the surface of the plate). The boundaries of the left and right droplet angle are marked in units of the scale division, n_1_ and n_2_, and then the parameter d is calculated by the difference n_2_ − n_1_. The drop height is determined similarly by turning the scale 90°. 

The value of cos θ was calculated according to:cos θ=(d2)2−h2(d2)2+h2.

#### 2.3.9. Porometry

The gas–liquid displacement porometer POROLUXTM 500 (Porometer, Belgium) was operated in the standard pressure range up to 35 bar. Porefil (Benelux Scientific, Nazareth, Belgium) was used as the wetting liquid with the lowest surface tension available on the market equal to 16 dyne/cm. The sample holder diameter was 25 mm, and data acquisition and analysis were carried out using LabView^®^-based software supplied by the manufacturer. The wet membrane sample was mounted in the sample holder after dropping out excess liquid adhered to the membrane surface.

#### 2.3.10. Permeability Measurements

The pure component permeances through the obtained POI-Gl-POSS membrane were measured according to the Daynes–Barrer technique [28,41,59,60,61,62] in a constant-volume variable-pressure apparatus for gas permeability measurements at the initial transmembrane pressure difference of 1.10 bar and at ambient temperature (about 298 K).

The experimental setup presented in Figure 1 was designed for high-precision single gas dead-end permeation tests. The setup comprises a typical permeation cell made of AISI316 stainless steel with PTFE sealing, a vacuum pump LVS 105 T-ef (Ilmvac, Germany), and a pumping station HiCube 80 Eco (Pfeiffer Vacuum, Germany). The initial vacuum generated in the system was 4 × 10^−5^ Pa. The membrane area was 2.5 × 10^−4^ m^2^. The feed- and permeate-side pressures were monitored by high-precision pressure sensors WIKA S-10 0-16 bar (Wika, Germany) having an accuracy of 0.5% of span (for the feed side), and MKS Baratron 750B 0–20 torr pressure transducer (MKS Instruments, USA) with an accuracy of 1% of reading (for the permeate side). Constant feed pressure was maintained at 110 kPa; the permeate pressure variation was recorded with a sampling rate of 10 ms. Each single-gas test was repeated at least three times at least 24 h apart before each run. As a permeation cell, a radial membrane module (position 1 on Figure 1) with a porous mechanical support [63,64] was used, which was vacuum degassed before each measurement. The samples area was 2.54 cm^2^. The leak level was determined by a pressure gauge after closing all valves before measurements. 

The single gas permeability values of hydrogen, nitrogen, and ammonia through each sample were measured for 3 gases (N_2_, H_2_, and NH_3_). After ammonia analysis, the permeability of hydrogen and nitrogen was tested again to show the chemical resistance of polymer. All experiments were carried out following safety precautions, which are specified in ISO/TC 58/SC 4 and ISO/TC 161. After the investigation of ammonia gas transport properties, the permeability coefficients of hydrogen and nitrogen did not change. 

The permeability coefficient *P* was calculated according to [65]:$$P=\frac{Vp_{2}T_{0}}{V_{m}P_{0}T}\frac{l}{S\tau \left(p_{1}-p_{2}\right)}.$$

Moreover, the Daynes–Barrer technique [28,41,59,60,61,62] gives us an opportunity to determine the diffusion coefficient (D). Furthermore, the sorption coefficient might also be easily calculated:$$D=\frac{l^{2}}{6\theta },$$
$$S=\frac{P}{D},$$
where *V*—downstream volume; m^3^; *V*_m_—molar volume m^3^/mol; *p*_2_—downstream pressure for each individual experiment, Pa; *p*_1_—upstream pressure, Pa; *P*_0_—atmospheric pressure, Pa; *T*—temperature, K; *T*_0_—273.15, K; *S*—membrane area, m^2^; *l*—membrane thickness, m, *τ*—time of experiment, s; and *θ*—time lag, s. Thus, the permeability coefficient was found in the International System of Units (SI, Le Système International d’Unités) of mol m m^−2^s^−1^Pa. The permeability coefficient was performed in Barrer (1 Barrer = 3.348 × 10^−16^ mol m m^−2^ s^−1^ Pa^−1^ = 10^−10^ cm^3^ (STP) cm cm^−2^ s^−1^ cm Hg^−1^). 

After permeability measurements, the ideal selectivity calculated as the ratio of the fast over the slow permeability coefficients for the following pair of gases: NH_3_/N_2_, NH_3_/H_2_. The choice of the gas pairs is stipulated by the necessity of the ammonia separation in the Haber Bosch process, which is an artificial nitrogen fixation process and the main industrial procedure for the production of ammonia today.

The permeability coefficient relative error for a single test is less than 5%; consequently, the selectivity relative error is 7.1%. The relative error of the measurement series was usually less than 15%. Taking into account the thickness measurement relative error of 2.4%, the diffusion coefficient relative error is 5.5% and the sorption coefficient relative error equals 7.5%.

Samples thickness was measured using the micrometer (MK 25, Russia) with a range of 0–25 mm with an error of ±5 µm. The polymeric film was covered with a flat glass plate of a known thickness to prevent direct contact of the measuring tip of the micrometer with the soft film. The thickness of the samples was 287.5, 298, 210, 234, 267, 314, and 283 µm in order of the Gl-POSS content increase in the samples.

#### 2.3.11. X-ray Analysis

X-ray powder diffraction (XRPD) measurements were performed on a Bruker D8 Advance diffractometer equipped with a Vario attachment and Vantec linear PSD, using Cu radiation (40 kV, 40 mA) monochromated by a curved Johansson monochromator (λ Cu K_α1_ 1.5406 Å). Room-temperature data were collected in the reflection mode. Samples in the form of rectangular fragments cut from the initial films 2.5 cm × 2.5 cm in size were placed in a standard sample holder, which was kept spinning (15 rpm) throughout the data collection. The diffraction patterns were recorded in the range of scattering angles 2θ 1°–70°, step 0.008°, with a step time 0.1–0.5 s.

Small angle X-ray scattering (SAXS) data for samples were collected with the Bruker AXS Nanostar SAXS system using CuKα (λ 1.5418Å) radiation from a 2.2 kW X-ray tube (40 kV, 35 mA) coupled with Gobbel mirrors optics and a HiStar 2D area detector. The beam was collimated using three pinholes with apertures of 800, 450, and 700 μm. The sample–detector distance was 64.2 cm and was controlled using an AgBh calibration sample. The range of available values of scattering angles was 0.1° < 2θ < 4.8°, which corresponds to the values of the wave vector equal to 0.007 Å^−1^ < s < 0.34 Å^−1^ (s = (4π/λ)sin(θ)). Scattering patterns were obtained for the samples at 23 °C in an evacuated chamber. The measurements were performed in transition mode with the use of samples in the form of films of a uniform thickness and 4 × 8 mm^2^ dimension. Samples were mounted on a standard sample holder and placed in an evacuated diffractometer chamber. For each sample, several experiments were performed, allowing control of the quality of the experiments. The 2-D scattering patterns were integrated using the SAXS program package [66]. The data were corrected for background scattering and were normalized to the scattering of the standard sample—glassy carbon. Calculation of structural parameters, simulation, and graphical representation of the results were performed using PRIMUS [67] and SASView [68] program packages.

## 3. Results and Discussion

### 3.1. POI-Gl-POSS Synthesis and Characterization

The opening of isocyanate groups of TDI initiated by a macroinitiator (PPEG) can proceed both at the C=O and at the N=C component (Scheme 1) [53,54,69], as shown in Scheme 1.

It was shown [53] that the possibility of a reaction leading to the predominant formation of O-polyisocyanate blocks or, conversely, polyisocyanurate structures is determined by the reaction conditions, such as the nature of the solvent, temperature, solvent content, and the use of cocatalysts. In this work, Gl-POSS was used to stabilize the terminal O-polyisocyanate units and to create framework structural elements in the polymer matrix. The molar excess of epoxy groups in Gl-POSS was taken, according to Table 1, in the range from 0.144 to 11.7. The most optimal conditions for the full involvement of epoxy groups in interaction with terminal O-polyisocyanate units and the appearance of frame structures were expected at a ratio of E/K = 0.73 and 1.44 ([Gl-POSS] = 0.5 and 1.0 wt%, respectively). With a further increase in the Gl-POSS content, the probability of the formation of frame structures rapidly decreases. 

The main process in the interaction is the polyaddition of TDI accompanied by the isocyanate groups’ opening of the carbonyl part and opening of the epoxide ring initiated by a terminal O-polyisocyanate link as shown by the proposed mechanism in Scheme 2. 

The chemical structures of POI-0 and POI-Gl-POSS were confirmed by FTIR analysis. Figure 2a,b shows that FTIR spectra have analytical low-intensity bands at 1710 and 1410 cm^−1^ due to the stretching vibrations of C=O bond of the isocyanurates. A shoulder at 1730 cm^−1^ is seen, characteristic for the C=O bond in the urethane group. 

The mole ratio [TDI] / [PPEG] = 15 as mentioned in the synthesis description procedure. However, only 1 mole of TDI is introduced into to the reaction of urethane formation, with the remaining 14 moles being used in the anionic process. Considering the low intensity of the bands at 1710 and 1410 cm^−1^ and the comparability of the band intensities at 1710 and 1730 cm^−1^, it can be concluded that only a small part of the isocyanate groups participates in isocyanurates and urethane formation processes (Figure 2b). Moreover, the main reaction is polyaddition, accompanied by the isocyanate groups’ opening of the carbonyl. Thus, the formed O-polyisocyanates do not contain a C=O bond. The shoulder at 1640–1680 cm^−1^ implies the presence of the N=C bond in O-polyisocyanates. It should be noted that the N=C bond has a very low intensity. For the samples prepared using Gl-POSS (Figure 2), the shoulder at 1670 cm^−1^ becomes more signified, and its intensity increases with the increasing Gl-POSS content. According to the literature [70], the band at 1620 cm^−1^, which corresponds to the carbonyl stretching vibrations in the urea, is also characterized by the low intensity. The intensity of this band in Gl-POSS polymers increases when the polyisocyanurates content reduces.

The fact that TDI cyclization has a negligible contribution to the overall reaction process is confirmed by the TGA analysis (Figure 3). Thus, for polyisocyanurates (ICN), the thermal-oxidative degradation onset temperature is in the range of 583 K, while the use of Gl-POSS leads to the lowering of the temperature to 523 K. The mass loss rate depends on the content of Gl-POSS and reflects a significant impact on the formation of the polymer matrix. 

### 3.2. Morphology Characterization of POI-Gl-POSS

The surface morphology of the obtained POI-Gl-POSS polymeric films was studied by AFM analysis (Figure 4). It was observed that there were some considerable clefts, which were determined as pores with a round shape. The visible granular structure is evidence of the features of the formation of a supramolecular structure.

The pore size was measured by the gas–liquid displacement porometer POROLUXTM 500 via the above-described conditions down to 15 nm. The measurements via AFM data were started from a first point when the downhill feature appeared [71]. The maximum pore size is less than 9 nm for the range of 0.1–2 wt% of Gl-POSS and less than 6 nm for the range of 5–8 wt% of Gl-POSS. 

### 3.3. Water Contact Angle

To confirm the results obtained based on AFM, we studied the surface properties of nanoporous polymers by measuring the distilled water contact angle. Figure 5 shows that the change of the Gl-POSS content in polymer has a considerable influence on the wettability of the polymer surface. The highest values of the water contact angle are observed for the polymers with the most regular supramolecular structure. The most likely reason of changes in the contact angle value in the range of the optimal Gl-POSS content is the formation of the cellular supramolecular structure as the voids surface is covered with flexible component. The content of hydrophobic propylene oxide blocks in PPEG is 70 wt%. At this, the terminal blocks of hydrophilic ethylene oxide are in contact with the polyisocyanate fragments. That is, when forming a hard frame, the flexible component fills the inner space of the frame with preferential localization of the polyoxypropylene component of PPEG on the voids surface. In the case when the content of Gl-POSS is lower and higher than the optimal value, it destroys the hierarchical supramolecular structure and leads to the corresponding changes in the surface-active properties of the polymer. 

To evaluate the features of the pore formation, the water absorption measurements of the polymer samples were carried out. It was important that the geometrical dimensions of the samples absorbed the ultimate amount of water, and remained unchanged. This means that the water fills the voids, taking up no place between the chains. For the POI-Gl-POSS-0 sample, the ultimate degree of water absorption is 14 wt%; for POI-Gl-POSS-0.5, it is 11 wt%; for POI-Gl-POSS-1, it is 8.5 wt%; for POI-Gl-POSS-2, POI-Gl-POSS-5, and POI-Gl-POSS-8, it is 6%; for POI-Gl-POSS-15, it is 10 wt%; and for POI-Gl-POSS-20, it is 12 wt%. Thus, with the Gl-POSS content in which the most regular supramolecular structure of a nanoporous polymer is formed, the lowest values of the rate and ultimate degree of the water absorption are observed. This reflects the thermodynamic problems faced by water molecules both in the primary sorption process and in their moving through the pore cavity coated with the hydrophobic polyoxypropylene blocks.

### 3.4. Thermal–Mechanical and Dynamic Mechanical Behavior of POI-Gl-POSS

The TMA and MLT study was performed for the investigation of the mechanical behavior of polymers at high temperatures, which destroy the intermolecular interactions in supramolecular structures, formed with the participation of the rigid component. The curves of the thermomechanical analysis and MLT study reflecting the complexity of the macromolecular and supramolecular organization of the studied polymers are shown in Figure 6 and Figure 7. 

The TMA curves show several relaxation processes that are affected significantly by the Gl-POSS content in the polymer. The first region covers the temperature range up to 373 K. Here, there is a small strain (up to 10 wt%) on the TMA curves due to the thermal effect on the segmental mobility of the flexible amphiphilic component.

The content of Gl-POSS has the greatest influence on the high–temperature ranges of relaxation processes. At 503 K, strain changes do not exceed 20–40%. After 513 K, deformation of the polymer is determined by the mass loss as a result of the thermal degradation. It should be noted that under the mechanical load, the sample obtained at 10 wt% of Gl-POSS has the highest thermal resistance and reaches 55% strain at 623 K.

The results indicate that the O-polyisocyanate blocks are the key element of the supramolecular organization of the studied polymers. Owing to this functionality, the domains are stable at high temperatures. Such stability of the domain structures suggests that urea, formed by the reaction of the isocyanate groups of the ortho-position in TDI, forms bridges between adjacent O-polyisocyanate units, leading to their stabilization and segregation due to both chemical and physical binding according to Scheme 3.

### 3.5. The Temperature Dependence of the Dielectric Loss Tangents of POI-Gl-POSS

The POI-0 and POI-Gl-POSS composites were characterized by the dielectric loss tangent (DLT) and mechanical loss tangent (MLT) to study their bulk properties. The use of Gl-POSS and the increase of its content amplify the resistance of POI to thermomechanical effects by forming framing structures. Due to the existence of O-polyisocyanates as a part of the rigid branched structure, the polymer matrix has spatially different architecture elements of rigid components. This fact is reflected in the temperature dependences of the MLT for POI-Gl-POSS. 

Figure 8a,b show the temperature dependence of the dielectric loss tangent for neat POI-0 and POI-Gl-POSS, respectively. In interpreting the DLT curves, it is taken into account that the flexible component (PPEG) is the amphiphilic block-copolymer of propylene and ethylene oxide. Therefore, the temperature dependences of the dielectric and mechanical loss tangents reflect not only the completeness of microphase separation of rigid and flexible chain components but also the separation of polyoxypropylene and polyoxyethylene blocks. Indeed, when the Gl-POSS content is in the range of 2 and 5 wt%, the behavior of the dielectric loss tangent curves markedly differs from neat POI-0 and POI-Gl-POSS-0.5, POI-Gl-POSS-10, POI-Gl-POSS-15, and POI-Gl-POSS-20. Thus, for the neat POI-0 and POI-Gl-POSS-0.5, two regions of α-transition (at 226 K and 246 K) are observed. The lowest glass transition temperature in the region of 226 K corresponds to the polyoxypropylene block.

The peak of the temperature dependence of DLT at 246 K is most likely by the α-transition of the polyoxyethylene block. For POI-Gl-POSS-2 and POI-Gl-POSS-5, the α-transition at 226 K becomes less pronounced, while at the range of POI-Gl-POSS from 2 to 5, the region of the α-transition at 246 K is observed as a well-separated peak. This suggests that under such conditions, the polyoxyethylene component is directly adjoined to the hard O-polyisocyanate unit and involved in the formation of spatial units of the polymer network and loses its ability to isolate. For POI-Gl-POSS-10, POI-Gl-POSS-15, and POI-Gl-POSS-20, the α-transition at 226÷231 K is more expressed and broad. The presence of only one peak in the α-transition region is caused by irregularity of the skeleton supramolecular structure at such high contents of Gl-POSS. As a result, the excellence of the microphase separation of polyoxypropylene and polyoxyethylene units is lost.

### 3.6. Mechanical Properties of POI-Gl-POSS

The static mechanical properties of the polymers were estimated by tensile tests at ambient temperature. The samples used for the tensile test were obtained in three areas of the Gl-POSS content (less than 1 wt%, 1–8 wt%, and more than 8 wt%) (Figure 9). Gl-POSS has a significant impact on the mechanical behavior of polymers. 

When the content of Gl-POSS is in the range from 1 wt% to 8 wt%, the obtained samples are high-modulus polymers and are not typical elastomers. The stress values at 7–8 wt% tensile strain depend on the content of Gl-POSS, and reached 60 MPa at 2 wt% of Gl-POSS. At less than 1 wt% and more than 8 wt% of Gl-POSS, the stress value at a tensile strain of 7%–8% drops significantly. The geometric dimensions of the samples returned to its original size within 2–5 min after the applied stress and the subsequent break. 

Thus, when the content of Gl-POSS is in the range from 1 wt% to 8 wt%, the tensile strength gets many times higher, accompanied by a sharp decrease in the elongation at breaking stress. The results were explained as follows. Gl-POSS is involved in the chemical reactions initiated by the active centers at the end of the hard O-polyisocyanate blocks. As a result, Gl-POSS serves as a hard and bulky branching agent. In the end, the formed O-polyisocyanate blocks make an association around Gl-POSS. 

In the case of the optimal Gl-POSS content, branched geometry is seen, and the domains of the O-polyisocyanate blocks create a rigid frame, having an inner volume greater than the volume of the flexible component, which is 50 wt%. As a result, the flexible component covers the surface of the geometrically aligned domains of hard segments, while the internal cavity of the frame is empty. A feature of such a supramolecular architecture is that the structure of the polymer becomes framing, and thereby flexible polyether components are located inside the cavities, preventing flexing in the process of uniaxial tension. Under these conditions, the polyoxypropylene component of the macroinitiator forms its own microphase outside the segregation of hard segments and Gl-POSS domains, while the polyoxyethylene component of the macroinitiator exists inside the area of formation of the spatial branched structures. 

The content of Gl-POSS below 1 wt% and higher than 8 wt% destroys the strict geometry of the frame. This results in the loss of the necessary geometry for the supramolecular structures and, thus, in deterioration of the physical and mechanical characteristics of the polymer with the Gl-POSS content below 1% and higher than 8% (Figure 9 and Figure 10). In this case, the flexible component forms its own microphase outside the segregation of hard segment domains and Gl-POSS.

### 3.7. X-ray Analysis

The X-ray powder diffraction data for the polymer samples with different Gl-POSS contents indicate the absence of crystalline phases in them and the amorphous state of the samples, which suggests that the association of POSS molecules into nanocrystallites does not occur.

Small-angle X-ray scattering of POI-Gl-POSS polymers of different compositions indicates microphase separation of the rigid and flexible components in these systems and the formation of ordered supramolecular systems of the paracrystalline type. In this case, the two-dimensional patterns correspond to the scattering from isotropic systems, as evidenced by a uniform intensity distribution around the primary X-ray beam (Figure 11). For comparison, the background scattering is also shown in the figure.

Figure 12 shows the one-dimensional small-angle scattering curves for all studied samples of POI-Gl-POSS polymers obtained by integrating two-dimensional patterns. An analysis of the scattering curves indicates a close character of the angular distribution of the intensity of small-angle X-ray scattering for all samples. This circumstance, in turn, may indicate the absence of significant structural changes in the samples.

The interference peaks indicate the structural ordering of the samples, i.e., about the availability of domains and their ordered arrangement in the volume of samples. The parameters of this ordering are fairly close for all the samples studied. The parameters of the paracrystalline lattice **d** in the studied samples (the distance between the centers of the domains) lie in the range of 85.3–82.0 Å. For the POI-Gl-POSS-5 polymer, the parameter **d** is noticeably smaller and equal to 79.1 Å. An estimate of the average sizes of the areas of paracrystalline ordering (Long range order) leads to a value of 273.1 Å for polymer POI-Gl-POSS-5, and to a range of 330.2–371.8 Å for the remaining samples.

A characteristic feature of the scattering pattern of sample with the highest Gl-POSS content (POI-Gl-POSS-15) is intense scattering in the region of the smallest scattering angles (Figure 11d), which is primarily due to the presence of large inhomogeneities whose sizes approach the upper limit of the values measured by this method (10–800 Å).

Taking into account the Babinet principle [72], with equal right, by these heterogeneities one can mean not only denser (compared to the matrix) domains but also void pores. However, in the case of the presence of such pores, the scattering from them should disappear in the case of a decrease in contrast when they are filled with something, for example, solvent or water. In order to test this hypothesis, all samples were kept in water for several hours and small-angle X-ray experiments were repeated with them, the results of which are shown in Figure 13 for two samples.

Three 2-D SAXS patterns are shown for each of the samples, which correspond to the initial state of the samples, the swollen state, and after 2 days of keeping the samples under room conditions. It can be noted that for the sample with the highest Gl-POSS content (POI-Gl-POSS-15), an irreversible disappearance of scattering is observed in the region of the smallest angles, which confirms to some extent the assumption of the presence of voids in this sample (Figure 13).

An analysis of the SAXS data of the samples obtained before swelling, after swelling in water and after settling in room conditions allowed us to confirm the conclusions based on the results of the physical and mechanical tests. So, typical for the samples is the skeleton of the supramolecular structure due to the presence of domains and bulk fragments connecting them. The absorption of water by the samples under study was carried out into voids and did not lead to disruption of the frame supramolecular structure due to the stiffness and the predominant contribution to its formation of chemical rather than intermolecular bonds.

### 3.8. Gas Separation Performance of POI-Gl-POSS Membranes

The obtained values for the POI-Gl-POSS membranes with different Gl-POSS contents at ambient temperature are presented in Table 2. As it was shown in [54,55] the POI-polymers without Gl-POSS have a cellular supramolecular structure, built of the globules of the flexible component and associates of the rigid O-polyisocyanate block. These samples have nanopores. According to Table 3, despite the large pore size, the sample is virtually impermeable for the nitrogen molecules, providing relatively high permeability values for ammonia. The POI-Gl-POSS samples were also permeable for the inert gases (N_2_, H_2_) and ammonia. Changes in the macromolecular and supramolecular structure are reflected in the permeability values of the inert gases and ammonia. 

The obtained diffusion coefficients are in good agreement with the H_2_ and NH_3_ molecules’ kinetic diameters, which are 2.89 and 2.6 Å [1], respectively, except the POI-Gl-POSS sample with a [Gl-POSS] content of 2.0 wt%. The H_2_ diffusion coefficients increase with an increase of the POI-Gl-POSS load in the samples; meanwhile, the NH_3_ diffusion coefficient remains the same (about of 3.8 m^2^ s^−1^), except POI-Gl-POSS 1 wt% with 5.9 m^2^ s^−1^. Nevertheless, the high ammonia permeability rate was mainly determined by its sorption rate in the studied samples, the value of which increases with the POI-Gl-POSS loading rate. A further increase in the POI-Gl-POSS content leads to a decrease in both the sorption and, as following, the permeability. The H_2_ sorption coefficients are comparatively low, due to the absence of an interaction with the membrane material. In light of the above, the high selectivity for the NH_3_/H_2_ system is determined mainly by the samples’ sorption ability. As for nitrogen, its diffusion coefficient should be smaller, because the kinetic diameter of the nitrogen molecule is 3.64 Å

The difference between the permeability of the control sample and the samples obtained at a very low Gl-POSS content (0.1–1.0 wt%) is a result of its participation in the reaction process, and shows a significant impact of the cooperative effects on the formation of the supramolecular structure of POI-Gl-POSS. Nitrogen molecules hardly pass through the polymers POI-Gl-POSS-0.1, POI-Gl-POSS-0.5, and POI-Gl-POSS-1.0, and the ammonia permeability increases with the growing Gl-POSS content.

The most likely reason is that in this case, the frame supramolecular structure is fragmentary, making the contact between the pores difficult. The frame pieces are alternated with the flexible chain component, which forms its own microphase outside the segregation of the hard segment domains and Gl-POSS. As a result, the supramolecular alignments alternate within the polymer, so that they form voids with the layers of the flexible component.

The non-polar H_2_ and N_2_ molecules do not penetrate through the flexible chain, and this explains their very low permeability as shown in Table 4 and Table 5. The high permeability value of ammonia, on the contrary, is due to its polarity and thermodynamic compatibility with the block-copolymers of ethylene oxide and propylene oxide as shown in Table 4 and Table 5. As a result, the ammonia molecules diffuse through the layer of the flexible-chain component. The proposed mechanism of high ammonia permeability is reflected in the absence of such side effects as membrane plasticization [27,28], which occurs in the increase–decrease cycle of the gas pressure. 

According to this study, the regular frame supramolecular structure is formed at 2–8 wt% of Gl-POSS. The inert gases’ permeability increases, but the ammonia permeability value slightly reduces in this range of the Gl-POSS content. The most likely reason of the revealed behavior is the alignment of the voids in a system of interconnected channels. 

Thus, the results indicate that ammonia is dissolved in PPEG. Hence, the supramolecular structural features of POI-Gl-POSS polymeric membranes introduced in this study result in a significant improvement of the separation efficiency of the ammonia-containing gas mixtures.

The performance, i.e., permeability, must be improved to be competitive with commercially available gas separation membranes. 

It was shown in a number of studies [26,73,74,75] that a polydimethylsiloxane-based membrane provides a high ammonia permeability (up to 12,000 Barrer) with a corresponding selectivity of 25.6 and 12.8 for NH_3_/N_2_ and NH_3_/H_2_ systems. Another one, among the well-studied polymers, poly(vinyltrimethylsilane), was characterized by an ammonia permeability of 830 Barrer and selectivity of 7.5 and 4.1 with regard to nitrogen and hydrogen, respectively [76]. The poly(trimethylsilylpropyne) membrane provides very high permeability for NH_3_ of 360,000 Barrer [76] and quite high selectivity for an NH_3_/H_2_ system of 50. Unfortunately, there is no data on nitrogen in this paper. The commercially available membrane Nafion 117 (Poly(perfluorosulfonate)) [77] demonstrates an NH_3_ permeability of 323,070 Barrer and very high selectivity for both NH_3_/N_2_ and NH_3_/H_2_ pairs of 702 and 452. 

The obtained results for POI-Gl-POSS membranes seem to be promising due to their high selectivity values for NH_3_/N_2_ and NH_3_/H_2_ systems. This fact may lead to the use of these membranes in an ammonia separation plant application as shown in Figure 14.

## 4. Conclusions

In summary, this paper described a novel nanoporous Gl-POSS-branched polymer synthesized by polyaddition of 2,4-toluene diisocyanate with macroinitiator, and using octaglycidyl polyhedral oligomeric silsesquioxane as a bulk branch node. It was found that the main process in the interaction is the polyaddition of 2,4-toluene diisocyanate accompanied by the isocyanate groups’ opening of the carbonyl part. The results suggest that the participation of Gl-POSS in the reaction with reactive terminal O-polyisocyanate links leads to the formation of the frame supramolecular structure. In the Gl-POSS content of 1–8 wt%, the supramolecular structure is the most ordered, and it fully reveals the segregated activity of the rigid component. Under these conditions, the polyoxypropylene component of the macroinitiator forms its own microphase outside the segregation of hard segments and Gl-POSS domains, while the polyoxyethylene component of the macroinitiator exists inside the area of the formation of the spatial branched structures. As a result, the polyoxypropylene component covers the surface of the geometrically aligned domains of hard segments, while the internal cavity of the frame is empty.

It was shown that the largest values of ideal selectivity towards the ammonia-gas system were achieved at the content of Gl-POSS in the amount of 0.5–1 wt% and higher than 2–8 wt% in the polymer. It has been suggested that the combination of high permeability with high selectivity for ammonia molecules is the result of alternating relatively large voids with the layers of block copolymers of ethylene oxide and propylene oxide, through which ammonia dissolves.

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
