# Peer review of "Synthesis and Characterization of Novel Nanoporous Gl-POSS-Branched Polymeric Gas Separation Membranes"

_membranes, 2020, doi:10.3390/membranes10050110_

Round 1

Reviewer 1 Report

This manuscript reports the synthesis of novel hybrid polymer and evaluation as membranes for ammonia separation.  The synthesis of the polymers proceeded by using a macro initiator (PPEG), using as Si source and as a curing agent octaglycidyl polyhedral oligomeric silsesquioxane and 2,4-toluene diisocyanate 151 98 % (TDI) as monomer. The films were obtained by casting method and characterize by common techniques used for materials science, TGA, FT-IR, AFM, dielectric and mechanical measurements, water contact angle, porometry and X-Ray) . The films obtained were tested as selective membranes using each pure gases H2, N2 and NH3

Tables, figures and sequence of the paper description are good, the results are interesting and to some extent, the conclusion is supported by the results presented. There are however weaknesses in the paper that are summarized below.

Overall decision: Major revision is needed before consideration of acceptance/publication. In the following, it was addressed general comments, after more specified ones.

Reviewer 2 Report

The paper entitled "Synthesis and characterization of novel nanoporous GL-POSS-branched polymeric gas separation membranes" by Zaripov I.I. et al. has been reviewed. The paper deals with the synthesis and characterization of new hybrid gas separation membranes. The elaboration of the results of experimental data is interesting and the discussion is held with the scientific method of approach. But there are a few questions to which the authors should reply before publication. To start with, there are some errors in the English language which should be eliminated (for example, line 25; line 27; line 136; line 306; line 506, etc.). Besides, the abbreviation for GL-POSS should be corrected – in some cases in is GL, but in others – it is Gl.

1.       The same temperature unit should be used throughout the text (°C or K).

2.       For clarity, the gas permeability should be given in Barrer.

3.       The FTIR peak values should be added on the spectra in Fig. 2. Besides, the authors compare the peak intensity. However, to do it the spectra normalization should be carried out. Is it the case?

4.       The measurement error should be added in Fig. 5 and 10.

5.       Line 474: it is Figure 8a and 8b, isn’t it?

6.       The temperature dependence of the dielectric loss tangent for hybrid films with GL-POSS higher than 8 % should be added in Fig. 8.

7.       FTIR spectrum of PPEG treated with pure ammonia should be given in the text.

Reviewer 3 Report

The paper of Zaripov et al discusses the synthesis and characterization of membranes based on Nanoporous POSS materials. The topic is of sure interest for the membrane community, but the paper needs major revision before publication.

  • Proof-reading is necessary in the entire text since several errors are present such as: “this polymers”; 10-4 the -4 superscript; in line 154 I think that cool should be cold, in several sections: Don’t= do not or does not.

  • In Materials and methods argon, helium and nitrogen are introduced, but the paper discusses hydrogen, nitrogen and ammonia.

  • Contact angle discussion is similar to a thesis or manuscript paragraph and not written for a scientific article. Please rephrase it in the same way of the other techniques.
  • What is the used membrane area? 2.5cm2 (line 283) or almost 13cm2 (line 290)?
  • Samples were measured 3 (line288) or 5 (line293) times?
  • The error of 15% was between the measurements or before and after ammonia exposure? In this case, possible changes due to ammonia
  • The authors should discuss better what they mean for pore-size in the discussion of pore-measurements and AFM since membranes for gas separation should be dense.
  • Line 412 “which findings”?
  • What about the errors in the contact angle measurements? Why the contact angle decrease from the neat polymer to Poss-0.5, then increase up toPoss-2.and then decrease again?
  • Figure 9, please, give a complete caption. For instance, for a and b mean? And give the same color to the same membrane when reported in two different graphs.
  • Line 592 to 599 why this introduction?
  • 600-601 this was already written in materials and methods. Several of these sentences can be deleted in the entire text to increase the readability.
  • Table 2, is there any reason why H2 was not detected for the neat polymer? It is a bit strange since it should be more permeable than N2.
  • Table 2, some errors seems to small “e.g. 120 +- 0.02”. Could you comment on them? What about the errors in the selectivities?
  • Why only 1% of POSS is capable to bring the N2 permeability from 1.9 (POSS-1) to 95 (POSS-2) Barrer? At the same time, H2 permeation decreases which is strange. Then at POSS-8, N2 and H2 have same values. Please, give also possible on why the polymer behaves like that.
  • Why NH3 increases up to POSS-1 and then decreases
  • Line 622 is partially true, the ammonia increases up to 1, then decreases, why? Also selectivity follows the same trend. Please, amend and discuss the trend
  • Link between the physical-chemical characterization and the gas transport properties is missing. For instance, why the dielectric loss tangent has been measured? How it can be correlated with the material acting as a membrane?
  • Comparison between the new data and literature data is missing. Are these polymers better with respect to others?
  • Considering that H2 and N2 are also measured, the authors could use H2/N2 Robeson upper bound to discuss the materials properties.

Round 2

Reviewer 1 Report

Dear authors.

Most of the suggestion made in the first round was made, but important descriptions are missed and is necessary to be improved for increasing the impact of this huge work made by the authors. Please take your time for considering a making good revision and avoid important mistakes.

Suggestions are summarized in the attachment document.

Overall decision: Minor revision is needed before consideration of acceptance/publication. 

Reviewer 3 Report

The authors replied to my comments, but this reviewer still concerns about the absence of H2 permeability in Table 2. Considering that H2 permeability is always higher with respect to N2 permeability, it should be detecteble. 
